# Toeplitz Operators on Fock Space Over $\mathbb{C}^n$ with Invariant Symbols under the Action of the Unit Circle

Carlos González-Flores [1,*], Luis Alfredo Dupont-García [2], Raquiel Rufino López-Martínez [2] and Francisco Gabriel Hérnandez-Zamora [2]

1    Escuela Superior de Ingeniería Mecánica y Eléctrica Zacatenco, Instituto Politécnico Nacional, Mexico City 07738, Mexico
2    Facultad de Matemáticas, Universidad Veracruzana, Veracruz 94294, Mexico; ldupont@uv.mx (L.A.D.-G.); ralopez@uv.mx (R.R.L.-M.); francischernandez@uv.mx (F.G.H.-Z.)
*    Correspondence: cfgonzalez@esimez.mx

**Abstract:** The first goal of this paper is to find a representation of the Fock space on $\mathbb{C}^n$ in terms of the weighted Bergman spaces of the projective spaces $\mathbb{CP}^{n-1}$; i.e., every function in the Fock space can be written as a direct sum of elements in weighted Bergman space on $\mathbb{CP}^{n-1}$. Also, we study the $C^*$ algebras generated by Toeplitz operators where the symbols are taken from the following two families of functions: Firstly, the symbols depend on the moment map associated with the unit circle, and secondly, the symbols are invariant under the same action. Moreover, we analyze the commutative relations between these algebras, and we apply these results to find new commutative Banach algebras generated by Toeplitz operators on Fock space of $\mathbb{C}^n$.

**Keywords:** Toeplitz operators; moment maps; $C^*$ algebras; commutative banach algebras; fock spaces

**MSC:** 30H20; 47L80; 32A25





## 1. Introduction

In recent years, an important subject of study has been the relationship between the $C^*$ algebras generated by Toeplitz operators and geometric tools such as Lie groups, representation theory, etc. Several authors found commutative $C^*$ algebras, which are generated by Toeplitz operators with invariant symbols under the action of some maximal abelian subgroup on different domains such as the unit disk, unit ball, Siegel domain, and projective spaces. For more details, see [1–7].

Another step in this direction was to find commutative $C^*$ algebras using geometry tools such as symplectic geometry and moment mapping; for more details, see [8–10]. Moreover, using these techniques, several authors found commutative Banach algebras generated by Toeplitz operators, which are not $C^*$; for example, see [11–18]. More precisely, they used quasi-homogeneous, quasi-radial, and some generalizations of type symbols, which are invariant under the action of some group $G$ or are associated with the moment map of $G$ on different domains.

On the other hand, several authors found representations of the spaces of the analytic functions in terms of analytic functions in domains of the lower dimension. For example, in [6,19] the Bergman space over the Siegel domain is decomposed as a direct integral of weighted Fock spaces. Following this approach, we found a representation of the Fock space of $\mathbb{C}^n$ in the function of the weighted Bergman spaces of the projective spaces $\mathbb{CP}^{n-1}$; i.e., every function in the Fock space can be written in terms of elements in the weighted Bergman space on $\mathbb{CP}^{n-1}$. In other words, we show that the Fock space on $\mathbb{C}^n$ is unitarily equivalent to the direct sum of the weighted Bergman spaces on $\mathbb{CP}^{n-1}$; i.e.,

$$U(F^2(\mathbb{C}^n))U^* = \bigoplus_{m \in \mathbb{Z}_+} \mathcal{A}_m^2(\mathbb{CP}^{n-1}),$$

where $U$ is unitary.

In this paper we take the action of $S^1 = \mathbb{T}$ on $\mathbb{C}^n$, defined by

$$t \cdot (z_1, \ldots z_n) \in S^1 \times \mathbb{C}^n \mapsto (tz_1, \ldots, tz_n) \in \mathbb{C}^n, \tag{1}$$

to study the $C^*$ algebras of the Toeplitz operators, considering the following cases:

(a). The symbols depend of the moment map associated to the action; i.e., every symbol has the form $a \circ \mu$ where $\mu$ is the moment map associated to the above action.

(b). The symbols are invariant under this action; i.e., every symbol $c$ satisfies the relation $c(\tau \cdot z) = c(z)$, for each $\tau \in S^1$ and $z \in \mathbb{C}^n$.

Using the above actions and their moment map, we introduce a coordinate system on $\mathbb{C}^n$, which is given by these geometrical objects; this coordinate system is very useful to find the representation of the Fock space $F^2(\mathbb{C}^n)$ in terms of the weighted Bergman spaces on $\mathbb{CP}^{n-1}$. Hence, the $C^*$ algebras generated by Toeplitz operators where the symbols are as in (a) and (b) are denoted by $\mathcal{A}$ and $\mathcal{B}$, respectively.

We show that each element of the algebra $\mathcal{A}$ is a direct sum of the multiples of the identity operator on each component of $\bigoplus_{m \in \mathbb{Z}_+} \mathcal{A}_m^2(\mathbb{CP}^{n-1})$. Similarly, we show that each element of the algebra $\mathcal{B}$ can be written as a direct sum of the Toeplitz operators on the weighted Bergman spaces of the projective space $\mathbb{CP}^{n-1}$. Using this result, we show the following relationship

$$T_a T_b = T_b T_a$$

where $T_a \in \mathcal{A}$ and $T_b \in \mathcal{B}$. Note that in [6,19] the authors presented a similar result for the Siegel domain.

Moreover, using the above relation between the algebras $\mathcal{A}$ and $\mathcal{B}$, we introduced a commutative Banach algebra of Toeplitz operators on the Fock space, which is obtained from $\mathcal{A}$ and the sub-algebra of $\mathcal{B}$ generated by Toeplitz operators with quasi-homogeneous symbols on the projective space (these symbols can be considered as a function of $\mathbb{C}^n$).

We have organized the rest of this paper in the following way: In Section 2, we present some known facts regarding the Bergman spaces and Toeplitz operators on projective space $\mathbb{CP}^{n-1}$. In Section 3, we present the action of $S^{n-1}$ on $\mathbb{C}^n$, along with the moment map and a coordinates system associated with this action. In Section 4, we present some known results about algebras generated by Toeplitz operators with quasi-radial and quasi-homogeneous symbols over the complex projective space. In Section 5, we present a connection between the space de Fock of $\mathbb{C}^n$ and the direct sum of the weighted Bergman spaces of $\mathbb{CP}^{n-1}$. Finally, in Section 6, we introduce some commutative Banach algebras of Toeplitz operators on the Fock space using the commutation relations between the algebras $\mathcal{A}$ and $\mathcal{B}$, presented in Section 4.

## 2. Preliminaries

As usual, the complex projective space $\mathbb{CP}^{n-1}$ is the complex $n-1$ dimensional manifold that consists of all elements $[w] = \mathbb{C}w \setminus \{0\}$, where $w \in \mathbb{C}^n \setminus \{0\}$. For every $j = 1, \ldots, n$ we have an open set

$$U_j = \{[w] \in \mathbb{CP}^{n-1} : w_j \neq 0\}$$

and a holomorphic chart $\varphi_j : U_j \to \mathbb{C}^{n-1}$ given by

$$\varphi_j([w]) = \frac{1}{w_j}(w_1, \ldots, \hat{w}_j, \ldots, w_n) = (z_1, \ldots, z_{n-1}). \tag{2}$$

where the notation $\hat{w}_j$ means that $w_j$ is omitted. The numbers $z_k$ are known as the homogeneous coordinates with respect to the map $\varphi_j$. Note that the collection of all such maps yields a holomorphic atlas of $\mathbb{CP}^{n-1}$.

From [2], we know that the volume element on $\mathbb{C}^{n-1}$ induced by the Fubini–Study metric has the following form

$$d\Omega = \frac{1}{(2\pi)^{n-1}} w^{n-1} = \frac{1}{\pi^{n-1}} \frac{dV(z)}{(1 + |z_1|^2 + \cdots + |z_{n-1}|^2)^{n-1}} \tag{3}$$

where $dV(z) = dx_1 \wedge dy_1 \wedge \cdots \wedge dx_{n-1} \wedge dy_{n-1}$ is the canonical Lebesgue measure on $\mathbb{C}^{n-1}$. Moreover, in polar coordinates we have that

$$dV(z) = rdr \prod_{k=1}^{n-1} \frac{dt_k}{it_k},$$

where $z = (t_1 r_1, \ldots, t_{n-1} r_{n-1}) \in \mathbb{C}^{n-1}$, $t = (t_1, \ldots, t_{n-1}) \in \mathbb{T}^{n-1}$ and $r = (r_1, \ldots, r_{n-1}) \in \mathbb{R}_+^{n-1}$. Also, the volume form for $\mathbb{C}^{n-1} = \mathbb{T}^{n-1} \times \mathbb{R}_+^{n-1}$ is given by

$$d\Omega = \frac{1}{\pi^n} \frac{rdr \prod_{k=1}^{n-1} \frac{dt_k}{it_k}}{(1 + r_1^2 + \cdots + r_{n-1}^2)^{n-1}},$$

where $rdr = r_1 dr_1 \cdots r_{n-1} dr_{n-1}$.

With respect to the coordinates induced by $\varphi_0$, given $m \in \mathbb{Z}_+$, the $m$-weighted measure on $\mathbb{CP}^{n-1}$ is defined by

$$\begin{aligned} dv_m(z) &= \frac{(n+m-1)!}{m!} \frac{d\Omega(z)}{(1 + |z_1|^2 + \cdots + |z_{n-1}|^2)^m} \\ &= \frac{(n+m-1)!}{\pi^{n-1} m!} \frac{dV(z)}{(1 + |z_1|^2 + \cdots + |z_{n-1}|^2)^{n+m}}. \end{aligned}$$

To simplify the notation, we use the same symbol $dv_m$ to denote the weighted measures for both $\mathbb{CP}^{n-1}$ and $\mathbb{C}^{n-1}$, respectively.

**Definition 1.** *The weighted Bergman space on $\mathbb{CP}^{n-1}$, with weight $m \in \mathbb{Z}_+$, is defined by*

$$\begin{aligned} \mathcal{A}_m^2(\mathbb{CP}^{n-1}) &= \{\zeta \in L^2(\mathbb{CP}^{n-1}, H^m) : \zeta \text{ is holomorphic}\} \\ &= \Gamma_{hol}(\mathbb{CP}^{n-1}, H^m), \end{aligned}$$

*where $H^m = H \otimes \cdots \otimes H$ is the tensor product of $m$ copies of $H = T^*$ and $T$ is the tautological or universal line bundle of $\mathbb{CP}^{n-1}$.*

It is also known that for every $m \in \mathbb{Z}_+$, the Bergman space $\mathcal{A}_m^2(\mathbb{CP}^{n-1})$ satisfies the following properties:

(i). With respect to the homogeneous coordinates of $\mathbb{CP}^{n-1}$, the Bergman space can be identified with the space $P^{(m)}(\mathbb{C}^{n-1})$ of all homogeneous polynomials of degree $m$ over $\mathbb{C}^{n-1}$.

(ii). The map $\Phi_0 : L^2(\mathbb{CP}^{n-1}, H^m) \to L^2(\mathbb{C}^{n-1}, v_m)$ defined by $\zeta \mapsto \hat{\zeta} = \zeta|_{U_0} \circ \varphi_0^{-1}$ is an isometry and it is well known that

$$\Phi_0(\mathcal{A}_m^2(\mathbb{CP}^{n-1})) = P_m(\mathbb{C}^{n-1}),$$

where $P_m(\mathbb{C}^{n-1})$ denotes the space of all polynomials on $\mathbb{C}^{n-1}$ of degree less than or equal to $m$.

Recall the following usual notation for multi-index: given $\alpha \in \mathbb{Z}_+^n$ and $z \in \mathbb{C}^n$, we have that

$$|\alpha| = \alpha_1 + \cdots + \alpha_n,$$

$$\alpha! = \alpha_1! \cdots \alpha_n!,$$

$$z^\alpha = z_1^{\alpha_1} \cdots z_n^{\alpha_n},$$

$$J_n(m) = \{\alpha \in \mathbb{Z}_+^n : |\alpha| \leq m\}.$$

Considering the identification of the Bergman space $\mathcal{A}_m^2(\mathbb{CP}^{n-1})$ with the space $P_m(\mathbb{C}^{n-1})$, we have that the monomial functions $z^{\alpha'} = z_1^{\alpha_1} \cdots z_{n-1}^{\alpha_{n-1}}$ form an orthogonal basis. Thus, the set of functions

$$\varphi_{\alpha'}(z) = \left( \frac{m!}{\alpha'!(m - |\alpha'|)!} \right)^{1/2} z^{\alpha'}, \quad \alpha' \in J_{n-1}(m) \tag{4}$$

is an orthonormal basis for $\mathcal{A}_m^2(\mathbb{CP}^{n-1})$, where the inner product is defined by

$$\langle f, g \rangle_m = \frac{(n+m-1)!}{\pi^{n-1} m!} \int_{\mathbb{C}^{n-1}} \frac{f(z)\overline{g(z)}dV(z)}{(1 + |z_1|^2 + \cdots + |z_{n-1}|^2)^{n+m}}$$

for all $f, g \in \mathcal{A}_m^2(\mathbb{CP}^{n-1})$. Furthermore, in local coordinates, the Bergman projection from $L^2(\mathbb{CP}^{n-1}, H^m)$ onto $\mathcal{A}_m^2(\mathbb{CP}^{n-1})$ is defined by

$$B_m(\psi)(z) = \frac{(n+m-1)!}{\pi^{n-1} m!} \int_{\mathbb{C}^{n-1}} \frac{\psi(w) K_m(z, w) dV(w)}{(1 + |w_1|^2 + \cdots + |w_{n-1}|^2)^{n+m}}$$

where $K_m(z, w) = (1 + z_1 \bar{w}_1 + \cdots + z_{n-1} \bar{w}_{n-1})^m$ and this function is called the Bergman kernel for $\mathcal{A}_m^2(\mathbb{CP}^{n-1})$.

**Definition 2.** *If* $a \in L^\infty(\mathbb{CP}^{n-1})$ *then the Toeplitz operator* $T_a$ *with symbol* $a$ *is the bounded operator on* $\mathcal{A}_m^2(\mathbb{CP}^{n-1})$ *defined by* $T_a^m(f) = B_m(af)$, *for each* $f \in \mathcal{A}_m^2(\mathbb{CP}^{n-1})$.

Note that the Toeplitz operator with symbol $a \in L^\infty(\mathbb{CP}^{n-1})$ can be represented as a matrix $A$ where the entries are given by

$$A_{\alpha', \beta'} = \langle T_a^m(\varphi_{\alpha'}), \varphi_{\beta'} \rangle_m = \frac{(n+m-1)!}{\pi^{n-1} m!} \int_{\mathbb{C}^{n-1}} \frac{a(z)\varphi_{\alpha'}(z)\overline{\varphi_{\beta'}(z)}dV(z)}{(1 + |z_1|^2 + \cdots + |z_{n-1}|^2)^{n+m}} \tag{5}$$

where $\alpha', \beta' \in J_{n-1}(m)$. This fact is clear since the Bergman space $\mathcal{A}_m^2(\mathbb{CP}^{n-1})$ is finite dimensional. For more detail, see [2,20].

## 3. Some Properties of the Action of $S^1$ on $\mathbb{C}^n$

Given a manifold $N$ and a Lie group $G$ with Lie algebra $\mathfrak{g}$ associated to $G$. If $G$ acts on $N$ then for every $X \in \mathfrak{g}$ we have a family of diffeomorphisms $\varphi_t^X(x) := (\exp tX) \cdot x$, which is an one parameter group and the vector field $X_N$ associated to this group $\{\varphi_t^X\}$; that is,

$$X_N(x) := \frac{d}{dt} \varphi_t^X(x)|_{t=0}.$$

It therefore makes sense to define $\rho\delta(X) = X_N$ . Unfortunately the map $\rho\delta$ is an anti-Lie algebra map:

$$[X, Y]_N = -[X_N, Y_N].$$

Recall the following definition:

- An action $\tau$ of $G$ on a symplectic manifold $(M, \omega)$ is symplectic if $\tau_g^* \omega = \omega$ for all $g \in G$.

- An action of a Lie group $G$ on a manifold $M$ is proper if the map $G \times M \to M \times M$ defined by $(g, m) \mapsto (g \cdot m, m)$ is proper. Recall that a continuous map $f : X \to Y$ between two topological spaces is proper if the preimage under $f$ of a compact set is compact.
- An action of a group $G$ on a set $X$ is free if for any $g \in G, x \in X$, the equation $g \cdot x = x$ implies that $g = 1$.

**Definition 3.** *Consider a Hamiltonian action of a Lie group $G$ on a symplectic manifold $(M, \omega)$. Let $\gamma : \mathfrak{g} \to (C^\infty(M), \{\cdot, \cdot\}), X \mapsto \varphi_X$ be a corresponding anti-Lie algebra map. The moment map $\Phi : M \to \mathfrak{g}^*$ corresponding to the action is defined by*

$$\langle \Phi(m), X \rangle = \varphi_X(m) = \gamma(X)(m)$$

*for $X \in \mathfrak{g}$ and $m \in M$, where $\langle \cdot, \cdot \rangle : \mathfrak{g}^* \times \mathfrak{g} \to \mathbb{R}$ is the canonical pairing.*

For more details about the above definition, we can see [21].

We recall that the standard action of the unit circle $\mathbb{T} = S^1$ over $\mathbb{C}^n$ is given by

$$\mathbb{T} \times \mathbb{C}^n \to \mathbb{C}^n \qquad (6)$$
$$(\tau, z) \mapsto (\tau z_1, \ldots, \tau z_n).$$

Also recall that the canonical symplectic form on $\mathbb{C}^n$ is defined by

$$\omega = i dz \wedge d\bar{z} = i \left( \sum_{j=1}^{n} dz_j \wedge d\bar{z}_j \right);$$

and we have that the action (6) is symplectic, proper, and free.

We denote by $\mathfrak{t}$ the Lie algebra associated to $\mathbb{T}$ which is given by scalar multiples of the identity matrix in $\mathfrak{u}(n)$; note that this algebra is generated by $\eta = (i, \ldots, i)$ and the dual algebra $\mathfrak{t}^*$ is generated by $\eta^*$. Now we consider an arbitrary element $X = \mu\eta \in \mathfrak{t}$ where $\mu \in \mathbb{R}$; thus we have a collection of diffeomorphisms over $\mathbb{C}^n$ defined by

$$\varphi_t^X(z) := (e^{i\mu t} \cdot z_1, \ldots, e^{i\mu t} \cdot z_n),$$

and the corresponding vector field to $X$ in polar coordinates is given by

$$X_{\mathbb{C}^n}(z_0) = \mu \sum_{j=1}^{n} \frac{\partial}{\partial \theta_j} |_{z=z_0},$$

where $z = (r_1, \ldots r_n, \theta_1 \ldots \theta_n)$. We calculate the contraction of the symplectic form $\omega$ with respect to $X_{\mathbb{C}^n}$

$$\omega(\cdot, X_{\mathbb{C}^n}) = \mu \sum_{j=1}^{n} r_j dr_j = d\left( \mu \sum_{j=1}^{n} r_j^2 \right) = d\left( \mu \|z\|^2 \right).$$

Therefore, we have that the moment map associated with the group $\mathbb{T}$ in the symplectic manifold $(\mathbb{C}^n, \omega)$ is given by

$$\Phi : \mathbb{C}^n \to \mathfrak{t}^*$$
$$z \mapsto \|z\|^2 \eta^*.$$

On the other hand, we apply the theorem of Marsden–Weinstein–Meyer to obtain a reduced space which is also a symplectic manifold. As a first step, we take the value

1, which is a regular value of the moment map $\Phi_k$, and then $\Phi_k^{-1}(1)$ is a submanifold of $\mathbb{C}^n$ and

$$\Phi_k^{-1}(1) = S^{2n-1},$$

where $S^{2n-1} = \{z \in \mathbb{C}^n : \|z\|^2 = 1\}$. In the second step, note that the action of $\mathbb{T}$ on $\Phi^{-1}(1)$ is free, thus

$$M_0 = \Phi^{-1}(1)/\mathbb{T} = S^{2n-1}/\mathbb{T}$$

is a smooth manifold, where the action of $\mathbb{T}$ on $S^{2n-1}$ is given by

$$\tau \cdot (z_1, \ldots, z_n) = (\tau \cdot z_1, \ldots, \tau \cdot z_n),$$

for all $\tau \in \mathbb{T}$ and $z \in S^{2n-1}$. It is immediate to see that

$$M_0 = S^{2n-1}/\mathbb{T} = \mathbb{CP}^{n-1}$$

and the orbit map

$$\pi : S^{2n-1} \to \mathbb{CP}^{n-1}$$
$$(z_1, \ldots, z_n) \mapsto ([z_1], \ldots, [z_n])$$

define a principal $\mathbb{T}$ bundle over $\mathbb{CP}^{n-1}$. Moreover, there exists a symplectic form $\omega_0$ on $\mathbb{CP}^{n-1}$ such that $\pi^*\omega_0 = \omega|_{\Phi^{-1}(1)}$.

Now, consider the local coordinates of the reduced manifold $\mathbb{CP}^{n-1}$ given by

$$\varphi_0 : \mathbb{C}^{n-1} \to \mathbb{CP}^{n-1}$$
$$w \mapsto [(1, w)],$$

where $(1, w) = (1, w_1, \ldots, w_{n-1}) \in \mathbb{C}^n$. We consider the embedding map $\tilde{\varphi}_0$ from $\mathbb{C}^{n-1}$ to $S^{2n-1}$ defined as follows:

$$\tilde{\varphi}_0(w) = \frac{1}{\rho}(1, w),$$

where

$$\rho = \|(1, w)\| = \sqrt{1 + |w_1|^2 + \cdots + |w_{n-1}|^2}. \tag{7}$$

The image $W_0 = \tilde{\varphi}_0(\mathbb{C}^{n-1})$ is a submanifold of $S^{2n-1}$ and we have that the symplectic form in local coordinates is given by

$$\omega_0 = \tilde{\varphi}_0^*(\omega|_{\Phi_k^{-1}(1)})$$

$$= i \frac{(1 + |z|^2)\sum_{j=1}^{n-1} dz_j \wedge d\bar{z}_j - \sum_{j,h=1}^{n-1} \bar{z}_j z_h dz_j \wedge d\bar{z}_h}{(1 + |z|^2)^2} = \omega,$$

where $\omega$ is the canonical symplectic form in $\mathbb{CP}^{n-1}$.

Consider the map

$$\Phi_0 : \mathbb{T} \times \mathbb{C}^{n-1} \to S^{2n-1}$$
$$(\tau, w) \mapsto \frac{\tau}{\rho}(1, w). \tag{8}$$

The image $\Phi_0(\mathbb{T} \times \mathbb{C}^{n-1})$ is a dense open set of $S^{2n-1}$, and the action of $\mathbb{T}$ is free, and we have that the map $\Phi_0$ is injective, since $\varphi_0$ is a local system of coordinates which is dense in $\mathbb{CP}^{n-1}$ (which is given by (2) with $j = 0$) and the quotient of $S^{2n-1}$ with $S^1$ is called the Hopf fibration, which is isomorphic to $\mathbb{CP}^{n-1}$. Therefore $(\mathbb{T} \times \mathbb{C}^{n-1}, \Phi_0)$ is a dense local

system of coordinates for $S^{2n-1}$, in similar way we can introduce $(\mathbb{T} \times \mathbb{C}^{n-1}, \Phi_j)$ associated to (2). Theses $(\mathbb{T} \times \mathbb{C}^{n-1}, \Phi_j)$, for $j = 0, \ldots, n-1$, provide a global system coordinates of $S^{2n-1}$. The volume element $dS$ on $S^{2n-1}$ can be expressed in the coordinates $(\mathbb{T} \times \mathbb{C}^{n-1}, \Phi_0)$ as follows:

$$dS = d\Omega \wedge \frac{d\tau}{i\tau}$$

where $d\Omega$ is the volume element of $\mathbb{CP}^{n-1}$ given in local coordinates by (3) and $\dfrac{d\tau}{i\tau}$ is the invariant volume of $\mathbb{T}$.

In summary, we have a system of coordinates of $\mathbb{C}^n$ given by

$$\begin{aligned} \Phi_0 : \mathbb{R}_+ \times \mathbb{T} \times \mathbb{C}^{n-1} &\to \mathbb{C}^n \\ (r, \tau, w) &\mapsto \frac{r\tau}{\rho}(1, w), \end{aligned} \tag{9}$$

where $\rho$ is given by (7).

## 4. Commutative Algebras Generated by Toeplitz Operators with Symbols in the Projective Space

In this section some basic definitions and concepts related to Toeplitz operators with symbols on the complex projective space are presented. For a more detailed description, we can see [14,15].

Let $k = (k_0, \ldots, k_\ell) \in \mathbb{Z}_+^\ell$ be a multi-index so that $|k| = n$. We will call such multi-index $k$ a partition of $n$. For the sake of definiteness, we will always assume that $k_0 \leq \cdots \leq k_\ell$. This partition provides a decomposition of the coordinates of $w \in \mathbb{C}^{n-1}$ as $w = (w_{(0)}, \ldots, w_{(\ell)})$, where

$$w_{(j)} = (w_{k_0 + \cdots + k_{j-1} + 1}, \ldots, w_{k_0 + \cdots + k_j}) = (w_{j,1}, \ldots, w_{j,k_j})$$

for every $j = 0, \ldots, \ell$, and the empty sum is 0 by convention. Each element $w \in \mathbb{C}^{n-1}$ has a decomposition as follows: For every $j = 0, \ldots, \ell$ we define $r_j = |w_{(j)}|$. And for any $j$ we write

$$w_{j,l} = r_j s_{j,l} t_{j,l} \quad \text{where } s_{j,l} = \frac{|w_{j,l}|}{r_j} \text{ and } t_{j,l} = \frac{w_{j,l}}{|w_{j,l}|}.$$

Correspondingly, we write $s_{(j)} = (s_{j,1}, \ldots, s_{j,k_j})$ and $t_{(j)} = (t_{j,1}, \ldots, t_{j,k_j})$. Note that $s_{(j)} = (s_{j,1}, \ldots, s_{j,k_j}) \in S_+^{k_j-1} := S^{k_j-1} \cap \mathbb{R}_+^{k_j}$, whenever $w_{(j)} \neq 0$ and $t_{(j)} = (t_{j,1}, \ldots, t_{j,k_j}) \in \mathbb{T}^{k_j}$ where $\mathbb{T}^r$ denote the $r$-dimensional torus.

**Definition 4.** *Let $k = (k_0, \ldots, k_\ell) \in \mathbb{Z}_+^\ell$ be a partition of $n$.*

(i). *A $k$-pseudo-homogeneous symbol is a function $\psi \in L^\infty(\mathbb{CP}^{n-1})$ that can be written in the form*

$$\psi([w]) = \mathbf{b}(s_{(0)}, \ldots, s_{(\ell)}) t^\alpha,$$

*where $\mathbf{b}(s_{(0)}, \ldots, s_{(\ell)}) \in L^\infty(\mathbb{S}_+^{k_0-1} \times \cdots \times \mathbb{S}_+^{k_\ell-1})$ and $\alpha \in \mathbb{Z}^n$ with $|\alpha| = 0$.*

(ii). *A $k$-quasi-radial symbol is a function $a \in L^\infty(\mathbb{CP}^{n-1})$ that can be written in the form*

$$\mathbf{a}([w]) = \tilde{a}(|w_{(0)}|, \ldots, |w_{(\ell)}|),$$

*for some function $\tilde{a} : [0, +\infty)^{\ell+1} \to \mathbb{C}$ which is homogeneous of degree 0.*

(iii). *A $k$-quasi-radial-pseudo-homogeneous symbol is a function in $L^\infty(\mathbb{CP}^{n-1})$ of the form $\mathbf{a}\,\mathbf{b}$ where $\mathbf{a}$ is $k$-quasi-radial symbol and $\mathbf{b}$ is a $k$-pseudo-homogeneous symbol.*

From Lemma 3.8 in [15], we know that for each $k$-quasi-radial symbol $\mathbf{a} \in L^\infty(\mathbb{C}^{n-1})$, the Toeplitz operator $T_{\mathbf{a}}$ acting on $\mathcal{A}_m^2(\mathbb{CP}^{n-1}) \simeq P_m(\mathbb{C}^{n-1})$ satisfies

$$T_{\mathbf{a}}(z^{\alpha}) = \gamma_{\mathbf{a},k,m}(\alpha)z^{\alpha},$$

for every $\alpha \in \mathbb{Z}_+^{\ell}$, where

$$
\begin{aligned}
\gamma_{\mathbf{a},k,m}(\alpha) \;=\; & \frac{(n+m-1)!}{(m-|\alpha|!)\prod_{j=1}^{\ell}(k_j-1+|\alpha_{(j)}|)!} \\
& \times \int_{\mathbb{R}_+^{\ell}} \frac{\mathbf{a}(\sqrt{r_1},\ldots,\sqrt{r_\ell})}{(1+r)^{n+m}} \prod_{j=1}^{\ell} r_{r_j}^{|\alpha_{(j)}|+k_j-1}\, dr_j.
\end{aligned}
\tag{10}
$$

Considering that a *k*-quasi-radial-pseudo-homogeneous symbol has the form **ab** where **a** and **b** are *k*-radial and *k*-pseudo homogeneous symbols, respectively. The previous result implies that the Toeplitz operator $T_{\mathbf{ab}}$, acting on $\mathcal{A}_m^2(\mathbb{CP}^{n-1}) \simeq P_m(\mathbb{C}^{n-1})$, satisfies the following relation:

$$
T_{\mathbf{ab}}(z^{\alpha}) = \begin{cases} \gamma_{\mathbf{ab},m}(\alpha)z^{\alpha+p} & \text{if } \alpha+p \in J_n(m) \\ 0, & \text{if } \alpha+p \notin J_n(m) \end{cases},
$$

for every $\alpha \in J_n(m)$, where

$$
\begin{aligned}
\gamma_{\mathbf{ab},m}(\alpha) = \; & \frac{(n+m-1)!}{(m-|\alpha|)!\prod_{j=1}^{\ell}\left(k_j-1+|\alpha_{(j)}|+\frac{1}{2}|p_{(j)}|\right)!} \\
& \times \int_{\mathbb{R}_+^{\ell}} \frac{\mathbf{a}(\sqrt{r_1},\ldots,\sqrt{r_\ell})\prod_{j=1}^{\ell} r_j^{|\alpha_{(j)}|+\frac{1}{2}|p_{(j)}|+k_j-1}}{(1+r_1+\cdots+r_{n-1})^{n+m}}\, dr_1\cdots dr_\ell \\
& \times \prod_{j=1}^{\ell}\left( \frac{(k_j-1+|\alpha_{(j)}|+\frac{1}{2}|p_{(j)}|)!}{\prod_{l=1}^{k_j}(\alpha_{j,l}+p_{j,l})!} \right. \\
& \left. \qquad \cdot \int_{\mathbb{S}_+^{k_j-1}} \mathbf{b}_j(\sqrt{s_{(j)}})\left[1-\sum_{l=1}^{k_j-1} s_{j,l}\right]^{\alpha_{j,k_j}+\frac{1}{2}p_{j,k_j}} \right. \\
& \left. \qquad \cdot \prod_{l=1}^{k_j-1} s_{j,l}^{\alpha_{j,l}+\frac{1}{2}p_{j,l}}\, ds_{j,l}. \right)
\end{aligned}
\tag{11}
$$

Moreover, if **ab** is a *k*-quasi-radial *k*-pseudo-homogeneous symbol with $|p_{(j)}| = 0$ for $j = 1,\ldots,\ell$, then the function $\gamma_{\mathbf{ab},m} : J_n(m) \to \mathbb{C}$ presented in (11) has the following form:

$$
\begin{aligned}
\gamma_{\mathbf{ab},m}(\alpha) = \; & \frac{(n+m-1)!}{(m-|\alpha|)!\prod_{j=1}^{\ell}\left(k_j-1+|\alpha_{(j)}|\right)!} \\
& \times \int_{\mathbb{R}_+^{\ell}} \frac{\mathbf{a}(\sqrt{r_1},\ldots,\sqrt{r_\ell})\prod_{j=1}^{\ell} r_j^{|\alpha_{(j)}|+k_j-1}}{(1+r_1+\cdots+r_{n-1})^{n+m}}\, dr_1\cdots dr_\ell \\
& \times \prod_{j=1}^{\ell}\left( \frac{(k_j-1+|\alpha_{(j)}|)!}{\prod_{l=1}^{k_j}(\alpha_{j,l}+p_{j,l})!} \right. \\
& \left. \qquad \cdot \int_{\mathbb{S}_+^{k_j-1}} \mathbf{b}_j(\sqrt{s_{(j)}})\left[1-\sum_{l=1}^{k_j-1} s_{j,l}\right]^{\alpha_{j,k_j}+\frac{1}{2}p_{j,k_j}} \right. \\
& \left. \qquad \cdot \prod_{l=1}^{k_j-1} s_{j,l}^{\alpha_{j,l}+\frac{1}{2}p_{j,l}}\, ds_{j,l}. \right)
\end{aligned}
\tag{12}
$$

for every $\alpha \in J_n(m)$ such that $\alpha + p \in J_n(m)$ and is zero otherwise. Also, we have the following decomposition

$$\gamma_{\mathbf{ab}} = \gamma_{\mathbf{a},k,m} \prod_{j=1}^{\ell} \gamma_{\mathbf{b}_j,k,P_{(j)}}. \tag{13}$$

So, the Toeplitz operators $T_{\mathbf{a}}$, $T_{\mathbf{b}_j t_{(j)}^{p_{(j)}}}$, $j = 1, \ldots, m$, pairwise commute and

$$T_{\mathbf{a} \prod_{j=1}^{\ell} \mathbf{b}_j t_{(j)}^{p_{(j)}}} = T_{\mathbf{a}} \prod_{j=1}^{\ell} T_{\mathbf{b}_j t_{(j)}^{p_{(j)}}}. \tag{14}$$

And therefore, the Banach algebra generated by this kind of Toeplitz operator is commutative. For more detail, see ([14] Section 3).

## 5. A Connection between the Space de Fock of $\mathbb{C}^n$ and the Direct Sum of the Weighted Bergman Spaces of $\mathbb{CP}^{n-1}$

The Fock or Segal–Bargmann space $F^2(\mathbb{C}^n)$ is defined by the set of all holomorphic functions on $\mathbb{C}^n$ satisfying the condition

$$\frac{1}{\pi^n} \int_{\mathbb{C}^n} |f(z)|^2 e^{-|z|^2} d\lambda(z) < \infty,$$

where $d\lambda(z)$ denotes the Lebesgue measure on $\mathbb{C}^n$. In this space, one can define an inner product as follows:

$$(f,g)_{F^2(\mathbb{C}^n)} = \frac{1}{\pi^n} \int_{\mathbb{C}^n} f(z)\overline{g(z)} e^{-|z|^2} d\lambda(z).$$

It is known that the Fock space $F^2(\mathbb{C}^n)$ is a Hilbert space with orthogonal basis $\{z^\alpha\}$ and $\|z^\alpha\|^2 = \alpha!$, where $\alpha \in \mathbb{Z}_+^n$. Then, the functions

$$\phi_\alpha(z) = \frac{z^\alpha}{(\alpha!)^{1/2}} \tag{15}$$

form an orthonormal basis for $F^2(\mathbb{C}^n)$.

Considering the above orthonormal basis, we define the following operator:

$$U : F^2(\mathbb{C}^n) \to \bigoplus_{m \in \mathbb{Z}_+} \mathcal{A}_m^2(\mathbb{CP}^{n-1}) \tag{16}$$

$$\phi_\alpha \mapsto \left\{ \delta_{m,|\alpha|} \phi_\alpha(1,w)(m!)^{1/2} \right\}_{m \in \mathbb{Z}_+} = \left\{ \delta_{m,|\alpha|} \varphi_{\alpha'} \right\}_{m \in \mathbb{Z}_+}$$

where $\alpha = (\alpha_1, \alpha') \in \mathbb{Z}_+^n$, and the functions $\phi_\alpha$ and $\varphi_{\alpha'}$ are defined by (15) and (4), respectively.

Note that in $\psi \in F^2(\mathbb{C}^n)$ we have that $\psi = \sum_{m \in \mathbb{Z}_+} \psi_m$, where $\psi_m$ is an homogeneous polynomial of degree $m$. Thus,

$$U(\psi) = \left\{ \psi_m(1,w)(m!)^{1/2} \right\}_{m \in \mathbb{Z}_+}$$

The corresponding adjoint operator $U^* : \bigoplus_{m \in \mathbb{Z}_+} \mathcal{A}_m^2(\mathbb{CP}^{n-1}) \to F^2(\mathbb{C}^n)$ has the form:

$$\left\{ \delta_{m,|\alpha|} \varphi_{\alpha'}(w) \right\}_{m \in \mathbb{Z}_+} \mapsto \frac{z_1^{m-|\alpha'|} \varphi_{\alpha'}(w)}{(m!)^{1/2}} = \phi_\alpha(z)$$

where $z = (z_1, w) \in \mathbb{C}^n$ and $\alpha = (m - |\alpha'|, \alpha') \in \mathbb{Z}_+^n$.

**Example 1.** *For $n = 2$, we have that $z = (z_1, z_2)$ and $\alpha = (\alpha_1, \alpha_2)$, thus*

$$\phi_{(\alpha_1, \alpha_2)}(z_1, z_2) = \frac{z_1^{\alpha_1} z_2^{\alpha_2}}{(\alpha_1! \alpha_2!)^{1/2}}.$$

*In consequence,*

$$U(\phi_{(\alpha_1, \alpha_2)}) = \left\{ \delta_{m, |\alpha|} \phi_{(\alpha_1, \alpha_2)}(1, w)(m!)^{1/2} \right\}_{m \in \mathbb{Z}_+} = \left\{ \delta_{m, |\alpha|} \frac{(m!)^{1/2} w^{\alpha_2}}{(\alpha_1! \alpha_2!)^{1/2}} \right\}_{m \in \mathbb{Z}_+}$$

*In particular, if $|\alpha| = \alpha_1 + \alpha_2 = m_0$ and $\varphi_{\alpha_2} \in \mathcal{A}_{m_0}^2(\mathbb{CP}^{n-1})$, then*

$$\varphi_{\alpha_2}(w) = \left( \frac{m_0!}{\alpha_2!(m_0 - \alpha_2)!} \right)^{1/2} w^{\alpha_2} = U^* \left( \left\{ \delta_{m, m_0} \frac{(m!)^{1/2} w^{\alpha_2}}{((m_0 - \alpha_2)! \alpha_2!)^{1/2}} \right\}_{m \in \mathbb{Z}_+} \right).$$

In summary, we have the following result, which is very important for the development of this work; i.e., we present a relation between the Fock space $F^2(\mathbb{C}^n)$ and the weighted Bergman spaces of $\mathbb{CP}^{n-1}$.

**Theorem 1.** *The operator $U$ maps $F^2(\mathbb{C}^n)$ onto*

$$\bigoplus_{m \in \mathbb{Z}_+} \mathcal{A}_m^2(\mathbb{CP}^{n-1}).$$

*Moreover, $U$ is an isometric isomorphism.*

**Proof.** By straightforward calculation, we have

$$(\phi_\alpha, \phi_\beta)_{F^2(\mathbb{C}^n)} = \langle U\phi_\alpha, U\phi_\beta \rangle_{\bigoplus_{m \in \mathbb{Z}_+} \mathcal{A}_m^2(\mathbb{CP}^{n-1})}.$$

The result follows from the previous equation and the fact that $\{\phi_\alpha\}$ is an orthonormal basis on the Fock space $F^2(\mathbb{C}^n)$. □

## 6. Toeplitz Operators on the Fock Space Over $\mathbb{C}^n$ with Invariant Symbols under the Action of $\mathbb{T}$

The aim of this section is to decompose a Toeplitz operator on the Fock space of $\mathbb{C}^n$ as a direct sum of Toeplitz operators on the weighted Bergman spaces of $\mathbb{CP}^{n-1}$ space using the unitary operator $U$, defined in the previous section.

Recall that a $\mathbb{T}$-invariant symbol $c$ on $\mathbb{C}^n$ is a function $c : \mathbb{C}^n \to \mathbb{C}$ which is invariant under the action of $\mathbb{T}$ given in Equation (6). In other words,

$$c(\tau \cdot z) = c(\tau \cdot z_1, \dots, \tau \cdot z_n) = c(z).$$

We denote by $\mathcal{C}$ the set of all $\mathbb{T}$-invariant functions. Each element $c \in \mathcal{C}$ has the form $c(z) = c(|z|, w)$, where $(r, \tau, w)$ are the local coordinates of $\mathbb{C}^n$ presented in (9).

In [6], the authors studied three types of invariant symbols under the action of a commutative subgroup. These symbols were considered as three types: The family $\mathcal{A}$, $\mathcal{B}$, and $\mathcal{C}$, respectively. The authors obtained that the Toeplitz operators with symbols in the family $\mathcal{A}$ are direct integrals of multiplication operators. In the family $\mathcal{B}$, the Toeplitz operators with symbols in this family are direct integrals of Toeplitz operators with the same symbol. And finally, for the family $\mathcal{C}$, the authors showed that the Toeplitz operators are a direct integral of Toeplitz operators, where the symbol depend of the base spaces of direct integral.

In [6], the authors studied three types of invariant symbols under the action of a commutative subgroup over the Siegel domain $D_n$. These symbols over the Bergman space of the Siegel domain were considered as three types:

- In the family $A$, a Toeplitz operator over the Bergman space of the Siegel domain is a direct integral of multiplication operators over the weighted Fock spaces.
- The family $B$, the Toeplitz operators over the Bergman space of $D_n$ with symbols in this family can be written as direct integrals of Toeplitz operators over the weighted Fock space, where the symbol is constant on each element of the direct integral.
- For the family $C$, the authors showed that the Toeplitz operators over the Bergman space of Siegel domain $D_n$ can be written as a direct integral of Toeplitz operators over the weighted Fock space, where the symbol varies on each element of the direct integral.

Furthermore, the authors decompose the weighted Bergman space of the Siegel domain as a direct integral of the weighted Fock space.

Inspired by the families presented in the previous paragraph, we will now consider three families of symbols over $\mathbb{C}^n$, which will be used to study Toeplitz operators on the Fock space $F^2(\mathbb{C}^n)$.

**Definition 5.** *We introduce the following families of functions:*

- $\mathbf{A} = \{a(|z|) : a \in L^\infty(\mathbb{R}_+)\} \subset L^\infty(\mathbb{C}^n)$;
- $\mathbf{B} = \{b(w) : b \in L^\infty(\mathbb{C}^{n-1})\} \subset L^\infty(\mathbb{C}^n)$;
- $\mathbf{C} = \{c(|z|, w) : c \in L^\infty(\mathbb{R}_+ \times \mathbb{C}^{n-1})\} \subset L^\infty(\mathbb{C}^n)$;

*where $(r, \tau, w)$ are the local coordinates given by (9); recall that $w$ is connected to the local coordinates of the projective space $\mathbb{CP}^{n-1}$.*

**Remark 1.** *Note that the symbols of the family* **A** *depend on the moment map associated with the action of the unit circle. In addition, the symbols of the family* **C** *are invariant under the action of the unit circle. Finally, the symbols of the family* **B** *are a special case of the symbols of the family* **C**.

The main result of this work is presented below, which connects the Toeplitz operators on the Fock space of $\mathbb{C}^n$ with the Toeplitz operators on the weighted Bergman space of the projective space $\mathbb{CP}^{n-1}$, since every Toeplitz operator on $F^2(\mathbb{C}^n)$ can be decomposed as a direct sum of Toeplitz operators on $\mathcal{A}_m^2(\mathbb{CP}^{n-1})$. Note that this result is analogous to Theorem 3.3 in [6].

**Theorem 2.** *Let $c$ be an element in* **C**; *the Toeplitz operator* $\mathrm{T}_c$ *acting on the Fock space $F^2(\mathbb{C}^n)$ is the unitary equivalent to the direct sum of Toeplitz operators $T_{c_m}^m$; that is,*

$$U\mathrm{T}_c U^* = \bigoplus_{m \in \mathbb{Z}_+} T_{c_m}^m \tag{17}$$

*where $U$ is given by (16) and $T_{c_m}^m$ is a Toeplitz operator acting on Bergman space with weight $m$ over $\mathbb{CP}^{n-1}$, with symbol*

$$c_m(w) = \frac{1}{(n+m-1)!} \int_{\mathbb{R}^s} c(r, w) e^{-r^2} r^{2m+2n-1} dr. \tag{18}$$

**Proof.** Given $\alpha, \beta \in \mathbb{Z}_+^n$, we have:

$$\left(\mathrm{T}_c \phi_\alpha, \phi_\beta\right)_{F^2(\mathbb{C}^n)} = \left(c(z) \frac{z^\alpha}{\sqrt{\alpha!}}, \frac{z^\beta}{\sqrt{\beta!}}\right)_{F^2(\mathbb{C}^n)}$$

$$= \frac{1}{\pi^n} \int_{\mathbb{C}^n} c(z) \frac{z^\alpha}{\sqrt{\alpha!}} \frac{\overline{z^\beta}}{\sqrt{\beta!}} e^{-|z|^2} d\lambda(z).$$

We use the coordinates $(r, \tau, w)$, which are associated with the moment map and symplectic reduction of the unit circle and defined by (9). Moreover, since the function $c$ is $\mathbb{T}$-invariant, we have the following relations:

$$
\left( T_c \phi_\alpha, \phi_\beta \right)_{F^2(\mathbb{C}^n)}
$$

$$
= \frac{1}{\pi^n} \int_{\mathbb{T}} \int_{\mathbb{R}_+} \int_{\mathbb{C}^{n-1}} c(r, w) \frac{\phi_\alpha(1, w)\overline{\phi_\beta(1, w)} e^{-r^2} r^{|\alpha|+|\beta|+2n-1} \tau^{|\alpha|-|\beta|}}{(1 + |w|^2)^{\frac{|\alpha|+|\beta|}{2}}} dr \, d\Omega(w) \frac{d\tau}{i\tau}
$$

$$
= \frac{1}{\pi^n} \int_{\mathbb{T}} \tau^{|\alpha|-|\beta|} \frac{d\tau}{i\tau} \int_{\mathbb{R}_+} \int_{\mathbb{C}^{n-1}} c(r, w) \frac{\phi_\alpha(1, w)\overline{\phi_\beta(1, w)} e^{-r^2} r^{|\alpha|+|\beta|+2n-1}}{(1 + |w|^2)^{\frac{|\alpha|+|\beta|}{2}}} dr \, d\Omega(w)
$$

$$
= \frac{\delta_{|\alpha|,|\beta|}}{\pi^{n-1}} \int_{\mathbb{R}_+} \int_{\mathbb{C}^{n-1}} 2c(r, w) \frac{\phi_\alpha(1, w)\overline{\phi_\beta(1, w)} e^{-r^2} r^{2|\alpha|+2n-1}}{(1 + |w|^2)^{|\alpha|}} dr \, d\Omega(w),
$$

where $d\Omega$ is given by (3).

On the other hand, for two elements $\{\delta_{k,m} \varphi_{\alpha'}(w)\}_{k \in \mathbb{Z}_+}$ and $\{\delta_{k,m'} \varphi_{\beta'}(w)\}_{k \in \mathbb{Z}_+}$ belonging to the space $\bigoplus_{m \in \mathbb{Z}_+} \mathcal{A}_m^2(\mathbb{CP}^{n-1})$, we have

$$
\left\langle U T_c U^* \{\delta_{k,m} \varphi_{\alpha'}(w)\}_{k \in \mathbb{Z}_+}, \{\delta_{k,m'} \varphi_{\beta'}(w)\}_{k \in \mathbb{Z}_+} \right\rangle_{\bigoplus_{m \in \mathbb{Z}_+} \mathcal{A}_m^2(\mathbb{CP}^{n-1})}
$$

$$
= \left( T_c U^* \{\delta_{k,m} \varphi_{\alpha'}(w)\}_{k \in \mathbb{Z}_+}, U^* \{\delta_{k,m'} \varphi_{\beta'}(w)\}_{k \in \mathbb{Z}_+} \right)_{F^2(\mathbb{C}^n)}
$$

$$
= \left( T_c \frac{z_1^{m-|\alpha'|} \varphi_{\alpha'}(w)}{(m!)^{1/2}}, \frac{z_1^{m'-|\beta'|} \varphi_{\beta'}(w)}{(m'!)^{1/2}} \right)_{F^2(\mathbb{C}^n)}
$$

$$
= \frac{\delta_{m,m'}}{\pi^{n-1}} \int_{\mathbb{R}_+} \int_{\mathbb{C}^{n-1}} 2c(r, w) \frac{\varphi_{\alpha'}(w)\overline{\varphi_{\beta'}(w)} e^{-r^2} r^{2m+2n-1}}{m!(1 + |w|^2)^m} dr \, d\Omega(w)
$$

$$
= \delta_{m,m'} \frac{(m+n-1)!}{\pi^{n-1} m!} \int_{\mathbb{C}^{n-1}} \frac{1}{(m+n-1)!} \int_{\mathbb{R}_+} 2c(r, w) e^{-r^2} r^{2m+2n-1} dr \frac{\varphi_{\alpha'}(w)\overline{\varphi_{\beta'}(w)}}{(1 + |w|^2)^m} d\Omega(w)
$$

And so, we define the following function:

$$
c_m(w) = \frac{1}{(n+m-1)!} \int_{\mathbb{R}_+} 2c(r, w) e^{-r^2} r^{2m+2n-1} dr,
$$

for all $m \in \mathbb{Z}_+$.

On the other hand, we obtain the following relations:

$$
\left\langle U T_c U^* \{\delta_{k,m} \varphi_{\alpha'}(w)\}_{k \in \mathbb{Z}_+}, \{\delta_{k,m'} \varphi_{\beta'}(w)\}_{k \in \mathbb{Z}_+} \right\rangle_{\bigoplus_{m \in \mathbb{Z}_+} \mathcal{A}_m^2(\mathbb{CP}^{n-1})}
$$

$$
= \delta_{m,m'} \frac{(m+n-1)!}{\pi^{n-1} m!} \int_{\mathbb{C}^{n-1}} c_m(w) \frac{\varphi_{\alpha'}(w)\overline{\varphi_{\beta'}(w)}}{(1 + |w|^2)^{m+n}} dV(w)
$$

$$
= \delta_{m,m'} \left\langle T_{c_m}^m(\varphi_{\alpha'}), \varphi_{\beta'} \right\rangle_m
$$

where $\alpha', \beta' \in J_{n-1}(m)$, and $\varphi_{\alpha'}, \varphi_{\beta'}$ are defined in Equation (4). Therefore, the result follows from Equation (5). $\square$

The following result describes a Toeplitz operator in Fock space, where the symbol depends on the moment map in terms of operators in complex projective space.

**Corollary 1.** *Given $a \in \mathbf{A}$, the Toeplitz operator $T_a$ acting on the Fock space $F^2(\mathbb{C}^n)$ is unitarily equivalent to the direct sum of multiplication operators $\gamma_a(m)I$; that is,*

$$
T_a = \bigoplus_{m \in \mathbb{Z}_+} \gamma_a(m) I
$$

*where $\gamma_a(m)$ is a function given by*

$$\gamma_a(m) = \frac{1}{(n+m-1)!}\int_{\mathbb{R}^s} a(r)e^{-r^2}r^{2m+2n-1}dr.$$

**Proof.** From Equation (18), we have that

$$a_m(w) = \frac{1}{(n+m-1)!}\int_{\mathbb{R}^s} a(r)e^{-r^2}r^{2m+2n-1}dr = \gamma_a(m).$$

From the above relation, we have that $\gamma_a(m)$ is a constant. In consequence, $T^m_{a_m} = \gamma_a(m)I$. Thus, the result follows from Equation (17).  □

**Corollary 2.** *Given $b \in \mathbf{B}$, the Toeplitz operator $\mathrm{T}_b$ acting in the Fock space $F^2(\mathbb{C}^n)$ is unitarily equivalent to the direct sum of Toeplitz operators $T^m_b$; that is,*

$$\mathrm{T}_b = \bigoplus_{m\in\mathbb{Z}_+} T^m_b$$

*where $T^m_b$ is a Toeplitz operator acting on the weighted Bergman space over $\mathbb{CP}^{n-1}$.*

**Proof.** We just need to calculate $b_m(w)$ has the following form:

$$\begin{aligned}
b_m(w) &= \frac{1}{(n+m-1)!}\int_{\mathbb{R}^\ell} b(w)e^{-r^2}r^{2m+2n-1}dr \\
&= b(w)\frac{1}{(n+m-1)!}\int_{\mathbb{R}^\ell} e^{-r^2}r^{2m+2n-1}dr = b(w).
\end{aligned}$$

In consequence, $T^m_{b_m} = T^m_b$.  □

For the symbols presented in the above statements, we can conclude the following results:

**Corollary 3.** *For any symbol $a = a(|z|) \in L^\infty(\mathbb{R}_+)$ and $b \in L^\infty(\mathbb{C}^{n-1})$, we have that $T_{ab} = T_aT_b = T_bT_a$, and*

$$U\mathrm{T}_{ab}U^* = \bigoplus_{m\in\mathbb{Z}_+} \gamma_a(m)T^m_b.$$

**Corollary 4.** *For any pair of symbols $a = a(|z|) \in L^\infty(\mathbb{R}_+)$ and $c = (|z|,w) \in L^\infty(\mathbb{R}_+ \times \mathbb{C}^{n-1})$, we have that $T_aT_c = T_cT_a$, and*

$$U\mathrm{T}_{ac}U^* = \bigoplus_{m\in\mathbb{Z}_+} \gamma_a(m)T^m_{c_m}.$$

**Remark 2.** *It is straightforward to check that, contrary to the case of Corollary 4, for the symbols of the previous corollary, we have that $T_aT_c \neq T_{ac}$, in general.*

The results of this section used the system coordinated (9). Note that every function in **B** is equivalent to a function in $\mathbb{CP}^{n-1}$ since it just depends on the variable $w$ in (9). Now, we consider functions $a \in \mathbf{A}$ and $\mathbf{a}, \mathbf{b} \in \mathbf{B}$, where $\mathbf{a}$ is a $k$-quasi-radial symbol and $\mathbf{b}$ is a $k$-quasi-homogeneous symbol given in Definition 4.

From Corollary 3 and Equation (14), the Toeplitz operators $T_a$, $T_\mathbf{a}T_\mathbf{b}$ pairwise commute and

$$T_aT_{\mathbf{ab}} = T_\mathbf{a}T_\mathbf{b}T_a. \tag{19}$$

## 7. Conclusions

In the present investigation, firstly we obtained the moment map $\Phi$ for the action of $S^1$ on $\mathbb{C}^n$ in a similar way as Sánchez and Quiroga in [10] obtained the moment map for the action on the unit ball of any maximal abelian subgroup of biholomorphisms of the unit ball. In consequence, Equations (8) and (9) provide a coordinate system, and this system uses the action of $S^1$ on $\mathbb{C}^{n-1}$ together with the function $\Phi$ and the symplectic reduction of $S^1$. These coordinates relate the space $\mathbb{C}^n$ with the moment map, the action of $S^1$, and the projective space $\mathbb{CP}^{n-1}$.

On the other hand, a crucial point of this paper was the introduction of the operator $U$ defined in (16), and we show that $U$ is unitary. This operator allows us to connect the Fock space of $\mathbb{C}^n$ with the weighted Bergman spaces of the complex projective space $\mathbb{CP}^{n-1}$.

Moreover, in Section 6, the operator $U$ was used to decompose a Toeplitz operator with symbols into one of the three families **A**, **B**, and **C** (presented in Definition 5 and contained in $L^\infty(\mathbb{C}^n)$) as direct sums of Toeplitz operators on the weighted Bergman spaces of the projective space $\mathbb{CP}^{n-1}$. And so, these decompositions are used to find commutativity relations between the algebras $\mathcal{A}$, $\mathcal{B}$, and $\mathcal{C}$ generated by the Toeplitz operators with symbols in the families **A**, **B**, and **C**, respectively.

On the other hand, as a future direction of this work, we will explore the use of the techniques used in this paper in several symmetric domains such as the unit ball, the Siegel domain, and Cartan domains, among others. In particular, we will use a Hamiltonian action and its moment map in these domains. Moreover, we will try to provide a characterization of the Bergman spaces in the mentioned domains. We will study the algebras generated by Toeplitz operators in these domains, where the symbol depends on the moment map or is invariant under the action of the group.

**Author Contributions:** Conceptualization, C.G.-F.; Investigation, C.G.-F., L.A.D.-G., R.R.L.-M., and F.G.H.-Z.; Writing—review and editing, C.G.-F., L.A.D.-G., R.R.L.-M., and F.G.H.-Z. All authors have read and agreed to the published version of the manuscript.

**Funding:** This research received no external funding.

**Data Availability Statement:** Not applicable.

**Acknowledgments:** The authors are appreciative of the reviewers who provided insightful criticism, suggestions, and counsel that helped them to modify and enhance the paper's final version.

**Conflicts of Interest:** On behalf of all authors, the corresponding author states that there are no conflict of interest.

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
