# Peer review of "Toeplitz Operators on Fock Space over with Invariant Symbols under the Action of the Unit Circle"

_axioms, doi:10.3390/axioms12121080_

Round 1
Reviewer 1 Report
Comments and Suggestions for Authors
Toeplitz Operators on Fock Space over Cn with invariant symbols under the action of unit the circle
The author need to put the conclusions to support the results.
Comments on the Quality of English LanguageToeplitz Operators on Fock Space over Cn with invariant symbols under the action of unit the circle
The author need to put the conclusions to support the results.
Author Response
First of all we thank you for your valuable and accurate comments, which will improve the quality of the manuscript. Below we will list the corrections that were made.
We added the conclusions to support the results.
Reviewer 2 Report
Comments and Suggestions for Authors
I have suggested the correction in the attached file.

Author Response
First of all we thank you for your valuable and accurate comments, which will improve the quality of the manuscript. Below we will list the corrections that were made.
Respect the question:
Thirdly, on page 6, in the last line, the authors give the mapping, which is injective. Is this mapping onto, or does the inverse of this mapping exist? Similar questions apply to all newly considered functions and mappings.
Answer:
The maping $\Phi_{0}$ ist’n sobrejective but the image is dense. Moreove, we provide a full of coordinate system of $S^{2n-1}$ using the coordinate system of the complex projective space.
Respect the question:
In Section 2 and Definition 3, what is meant by the symbol g?
Answer:
We specify the symbol $g$.
We added the conclusions to support the results.
Reviewer 3 Report
Comments and Suggestions for Authors
The results are new and original, and could be used by those interstate in this area of study.
The main strength of the article are:
1. The results are new and original;
2. The results and their proofs are correct;
3. The citations of the "References" are well choose, without inappropriate self citations;
4. The complexity of theresults and their proofs is astrong point for publish this submission in the Special Issue.
Since I don't have any objections, I recommend to publish the paper in the present form.
Author Response

(The authors gave the same response as above.)

Reviewer 4 Report
Comments and Suggestions for Authors
attached

Author Response
First of all we thank you for your valuable and accurate comments, which will improve the quality of the manuscript. Below we will list the corrections that were made.
We added the Section 5, this change was introduced to improve the structure the paper.
We added some coments at begining of the sections 3 and 5 to improve readability.
In the new Section 4, we emphasized the importance of the $U$ operator.
In the new Section 5, we present similar work with respect to Toeplitz operators on the Siegel domain, which inspired our work.
In the Section 1, we fixed the confution with the Bergman kernel of the complex projective space.
We fixed all the minor bugs.
We added the conclutions.
Round 2
Reviewer 2 Report
Comments and Suggestions for Authors
The quality is improved as compared to the old version of the manuscript.
Authors repated the "Preliminaries" sections as first in page 4 and then on page 5, so correct it.
Authors are suggested to include at least one sentence of future directions of their work in conclusion section.
Author Response
Authors repated the "Preliminaries" sections as first in page 4 and then on page 5, so correct it.
We review all sections in this paper that are:
0. Introduction (page 1 to 2)
1. Preliminaries. (page 2 to 4)
2. Some Properties of the Action of $S^1$ on $C^n$. (page 4 to 7)
3. Commutative Algebras Generated by Toeplitz Operators with Symbols in the Projective Space. (page 7 to 9)
4. A connection between the space de Fock of C n and the direct sum of the weighted
Bergman spaces of $CP^{n−1}$ . (page 9 to 11)
5. Toeplitz Operators on the Fock Space over C n with invariant symbols under the action of $T$. (page 11 to 14)
6. Conclusions. (page 14)
For this reason, we apologize that we haven't found the proposal you gave us. Perhaps there was some confusion with a previous version.
Authors are suggested to include at least one sentence of future directions of their work in conclusion section.
We add the following sentence in Conclutions:
On the other hand, as a future direction of this work, we will explore the use of the techniques used in this paper in several symmetric domains such as the unit ball, the Siegel domain, Cartan domains among others. In particular, we will use a Hamiltonian action and its moment map in these domains. Moreover, we will try to provide a characterization of the Bergman spaces in the mentioned domains. We will study the algebras generated by Toeplitz operators in these domains, where the symbol depends on the moment map or is invariant under the action of the group.
All the authors of this article would like to thank you for your comments and suggestions that helped us improve the presentation and content of this article.
Reviewer 4 Report
Comments and Suggestions for Authors
The revised article contains additional argumentations and related calculations which make the results obtained by authors much more readable and suitably motivated. There have been also more clearly specified authors original results, modulo those obtained by others. The latter still should be slightly supplemented by authors. Summarizing, all that makes the presented manuscript suitable for publication.
Author Response
All the authors of this article would like to thank you for your comments and suggestions that helped us improve the presentation and content of this article.